# Impact of the “Sling Shot” Supportive Device on Upper-Body Neuromuscular Activity during the Bench Press Exercise

**DOI:** 10.3390/ijerph17207695

**Published:** 2020-10-21

**Authors:** Grzegorz Wojdala, Artur Golas, Michal Krzysztofik, Robert George Lockie, Robert Roczniok, Adam Zajac, Michal Wilk

**Affiliations:** 1Institute of Sport Sciences, The Jerzy Kukuczka Academy of Physical Education in Katowice, 40-065 Katowice, Poland; a.golas@awf.katowice.pl (A.G.); m.krzysztofik@awf.katowice.pl (M.K.); r.roczniok@awf.katowice.pl (R.R.); a.zajac@awf.katowice.pl (A.Z.); m.wilk@awf.katowice.pl (M.W.); 2Department of Kinesiology, California State University, Fullerton, CA 92831, USA; rlockie@fullerton.edu

**Keywords:** resistance training, EMG, internal movement structure, training equipment, powerlifting gear

## Abstract

The aim of this study was to compare the muscle activity between the sling shot assisted (SS) and control (CONT) flat barbell bench press for selected external loads of 70%, 85%, 100% one-repetition maximum (1RM). Ten resistance-trained men participated in the study (age = 22.2 ± 1.9 years, body mass = 88.7 ± 11.2 kg, body height = 179.5 ± 4.1, 1RM in the bench press = 127.25 ± 25.86 kg, and strength training experience = 6 ± 2.5 years). Evaluation of peak muscle activity of the dominant body side was carried out using surface electromyography (sEMG) recorded for the triceps brachii, pectoralis major, and anterior deltoid during each attempt. The three-way repeated measure ANOVA revealed statistically significant main interaction for condition x muscle group (*p* < 0.01; η^2^ = 0.569); load x muscle group (*p* < 0.01; η^2^ = 0.709); and condition x load (*p* < 0.01; η^2^ = 0.418). A main effect was also observed for condition (*p* < 0.01; η^2^ = 0.968); load (*p* < 0.01; η^2^ = 0.976); and muscle group (*p* < 0.01; η^2^ = 0.977). The post hoc analysis for the main effect of the condition indicated statistically significant decrease in %MVIC for the SS compared to CONT condition (74.9 vs. 88.9%MVIC; *p* < 0.01; ES = 0.39). The results of this study showed that using the SS significantly affects the muscle activity pattern of the flat bench press and results in its acute decrease in comparison to an equal load under CONT conditions. The SS device may be an effective tool both in rehabilitation and strength training protocols by increasing stability with a reduction of muscular activity of the prime movers.

## 1. Introduction

The bench press is one of the three competitive lifts performed in powerlifting. It is also an individual sport discipline, with annual World and European Championships being organized (IPF, 2019). The bench press is also a basic resistance exercise for the upper body [1,2,3]. Depending on the structure of the movement, muscles with the highest involvement during the bench press movement include the anterior deltoid, pectoralis major, and triceps brachii [4,5,6,7,8]. According to Krol and Golas [9], the pectoralis major is the prime mover, while the anterior deltoid and triceps brachii act as supportive prime movers. It is worth noting that the external load alters the change in the pattern of muscle activity; for example, at maximal load, the pectoralis major acts as the supportive prime mover while the anterior deltoid becomes the prime mover [9]. Moreover, it should be taken into consideration that muscle activity may change depending on the bench press technique modifications or application of variable resistance [10,11]. Recruitment of motor units and the frequency of stimulations increase in parallel with the increase in external load, resulting in the achievement of the desired muscle tension and greater force [9]. In addition, the tempo of movement can also affect the muscular activity [12]. Lehman [5] found that faster movement tempo results in higher total activation of the muscles involved in the movement. Surface electromyography (sEMG) is a method of recording and analyzing electric signals generated by active muscles [13]; sEMG measurements provide the opportunity to thoroughly examine muscular activity, cooperation with other elements of the musculoskeletal system, and to evaluate the effects of fatigue [2].

While there is extensive literature on the bench press movement [2,5,9,14,15], far less attention has been given to the influence of performance enhancing equipment on maximum strength and muscle activity. Currently, powerlifters at the highest level are implementing specialized gear (i.e., squat suits, bench press shirts, deadlift suits, and knee wraps) to enhance performance [16,17]. Research suggests that bench press shirts alter the mechanics of the bench press movement through the “rebound” effect during the positive work [17], allowing greater loads to be lifted [18]. An innovative alternative to the bench press shirt that can be implemented into training includes the sling shot (SS) [19]. The SS is an upper body device made of elastic material with two sleeves that wraps around the elbows. While lowering the barbell to the chest during the bench press, the SS assists in movement by stretching the material and creating elastic tension. According to the manufacturer, the SS allows use of 10–15% more weight in pressing movements while reducing the tension on shoulders and elbows. Anecdotal evidence suggests that the SS is most often used as an overloading tool for the bench press through a full range of motion. Considering the construction and usage of bench press shirts, the mechanisms mentioned above can be equally effective using the SS. This device allows utilization of the elastic recoil effect, which is used by athletes to overcome greater loads or to perform more repetitions at a certain load [20,21]. Furthermore, the SS device may be an effective tool for reducing muscle activity and increasing stability, which may allow training through a full range of motion during minor injuries and support the rehabilitation process, ensuring a faster return to full load bench press training [8,22]. Moreover, due to the lower tightness compared to bench press shirts, it allows the athlete to familiarize himself with the use of specialized equipment. Therefore, the SS seems to enhance one-repetition maximum (1RM) bench press performance, despite the decreased activity of some of the muscles involved in this movement. Despite the reduction of the physiological stimulus, the use of the SS may be beneficial in building self-confidence and getting familiar with supramaximal loads. Additionally, sEMG measurements indicate decreased muscle activity of the prime movers while wearing the SS on submaximal loads allowing to overload a specific phase of the movement [23]. Through increases in elastic energy, the SS creates more favorable biomechanical conditions for generating greater initial mean and peak velocity of the barbell during exercises such as the bench press [20].

To the authors’ knowledge, there are only two studies investigating the impact of the SS on bench press performance and sEMG [20,23]. However, recent studies have focused on analyzing loads of 100% 1RM or greater [23] on multiple repetitions under significant fatigue [20,21]. We attempted to examine the acute effects of the SS during the bench press on the neuromuscular activity of the anterior deltoid, pectoralis major, and triceps brachii to evaluate how they differ depending on the external load applied. Moreover, apart from the maximum load, the measurements were also made on selected submaximal loads considered as typical training loads among resistance trained men, which allow performance of the exercises with appropriate technique [8]. Bearing in mind that the use of the SS is focused on muscle strength and power enhancement and the impact of its use on muscle activity during repetitions performed to failure [8,24,25], the emphasis was placed on low fatigue by performing a 1RM test on separate days and limiting the number of repetitions, which is what distinguishes this study from previous ones. The lack of progressive fatigue within the sets is commonly used in muscle strength and power development programs [26]. Our initial hypothesis states that the SS elastic device and the external load have a significant impact on the sEMG of the prime movers during the flat bench press. We hypothesize that the muscle activity of the prime movers will be significantly reduced during the bench press using the SS device with the same loads, regardless of fatigue due to the elastic assistance of the device in overcoming the external load.

## 2. Materials and Methods

### 2.1. Participants

The study involved 10 men (age = 22.2 ± 1.9 years, body mass = 88.7 ± 11.2 kg, body height 179.5 ± 4.1, and 1RM in the bench press = 127.25 ± 25.86 kg) with a minimum of three years of resistance training experience (6 ± 2.5 years). All of the participants were right-handed. The participants were allowed to withdraw from the experiment at any moment, and were free of any injuries. All study participants were informed about the benefits and potential risks of the study before providing their written informed consent for participation. The research was approved by the Ethics Committee of The Jerzy Kukuczka Academy of Physical Education in Katowice (10/2018) and executed according to the ethical standards of the latest version of the Declaration of Helsinki, 2013.

### 2.2. Study Design and Procedure

The experiment was conducted following a randomized crossover design. Each subject performed two experimental sessions: one with the sling shot elastic device during the bench press protocol (SS) and one control test protocol without it (CONT), preceded by one familiarization session. One week before the main experiment, a familiarization session preceded the 1RM in the free-weight flat bench press. The 1RM test was conducted in accordance with the guidelines based on the most recent bench press exercise literature to ensure reliability and validity [27]. Experimental sessions consisted of performing the bench press alternatively with or without the SS in a random order with progressive loads (70%, 85%, and 100% 1RM) to record peak muscle activity of the anterior deltoid, pectoralis major, and triceps brachii. The entire research procedure lasted three weeks, with a one week interval between each trial (Figure 1). Study participants were required to refrain from resistance training 72 h prior to each experimental session, and were familiarized with the exercise protocol. All testing sessions were performed at the same time of the day for each participant and between participants in the Strength and Power Laboratory at the Academy of Physical Education in Katowice.

### 2.3. Familiarization Session and One-Repetition Maximum Test

Prior to the 1RM trials, the participants were re-evaluated for their technical execution of the exercise with and without the SS. During the familiarization session the SS, size was adopted for each participant on the basis of bodyweight and consultation with a resistance training coach (medium, large, and extra-large size range, whereas each size provides the same tension), and the bench press grip width adopted for all sessions was 150% bi-acromial distance of each subject [15]. The participants arrived in the laboratory at the same time of the day as the upcoming experimental sessions and cycled on an ergometer for 5 min, followed by a general upper body warm-up. Next, the participants performed a specific warm-up consisting of 15, 10, and 5 bench press repetitions using 20%, 40%, and 60% of their estimated 1RM [7,28,29]. Estimation of 1RM was only used during the familiarization session to determine the actual 1RM and referred to free-weight flat bench press. For evaluation of actual 1RM, the loading started at 70% estimated 1RM and was increased by 2.5 to 10 kg for each subsequent attempt, and the process was repeated until failure [27,30,31,32,33]. In total, each participant performed between four and six attempts in the main session. The rest of the intervals between successive trials lasted 5 min [9]; the participants were verbally motivated to make a maximum effort. Three spotters were present during all attempts to ensure safety and technical proficiency [7]. The participants executed single repetitions in each subsequent set using a 2/0/V/0 tempo of movement, which denotes a 2 s negative work during lowering of the barbell, no pause during the transition phase, and a volitional movement tempo during the positive work associated with upward displacement of the barbell [12,28,34]. All repetitions were performed without bouncing the bar off of the chest and without raising the hips off of the bench [27,29].

### 2.4. Experimental Session

The main experimental protocols were conducted at the same time of the day to avoid the effects of circadian rhythm on bench press results. Two testing sessions (CONT and SS) were used for the experimental trials and completed by each participant. The same warm-up protocol was adopted, and an identical rack height and grip width was implemented. All participants performed single bench press attempts alternatively with or without the SS (Figure 2), with the external loads (70%, 85%, and 100% 1RM evaluated during familiarization session), and 5 min rest intervals were allowed between successive attempts. Following the seven-day interval, the participants returned to the laboratory and completed consecutive trials.

### 2.5. Electromyography

An eight-channel Noraxon TeleMyo 2400 system (Noraxon USA Inc., Scottsdale, AZ, USA; 1500 Hz) was used for recording and analysis of biopotentials from the muscles. The sEMG was recorded for three muscles: anterior deltoid, pectoralis major, and triceps brachii on the dominant side of the body. Before placing the gel-coated self-adhesive electrodes (Dri-Stick Silver circular sEMG Electrodes AE-131, NeuroDyne Medical, Cambridge, MA, USA), the skin was shaved, abraded, and washed with alcohol. The electrodes (11 mm contact diameter and a 2 cm center-to-center distance) were placed along the presumed direction of the underlying muscle fiber according to the recommendations by SENIAM [35]. The grounding electrode was placed on the connection with the anterior deltoid muscle. Landmarks to place the electrodes were used to ensure repeatability of the mounting location. The EMG signal was recorded at a sampling frequency of 1000 Hz. Signals were band pass filtered (8–450 Hz), after which EMG signals were converted into root mean square (RMS). Video recording was used only for identification of the beginning and completion of the movement without determining phases of the bench press. Two to three second tests of isometric contractions were performed before and after completion of all the tests in a single day in order to normalize sEMG records, according to the SENIAM procedure [35]. The maximum voluntary isometric contraction (MVIC) positions were selected according to standardized procedures, which were selected based on frequently used muscle test positions for the prime movers important during the bench press exercise movements [2]. Importantly, previous studies have shown high reliability of MVIC measurements (intraclass correlation coefficient ~0.98) [36]. Participants were instructed to gradually increase the force of the muscle contraction over a period of 2 s, and then hold to the MVIC for another 3 s. The triceps brachii MVIC values were recorded with the anterior deltoid MVIC during a seated shoulder abduction with 90° arm flexion, the seated triceps extension with the elbow flexed to 90°, and the pectoralis major MVIC during an isometric smith machine bench press immobilized by supramaximal weight at 90° elbow flexion [35]. The highest value was selected from the entire motion of one repetition of bench press for further analysis (to estimate peak maximum voluntary isometric contraction values, MVIC, %).

### 2.6. Statistical Analysis

Data were presented as means ± standard deviations. Entire calculations were performed using Statistica 9.1 (Hillview, Palo Alto, CA, USA) software. All variables showed a normal distribution, conforming to the Shapiro–Wilk test. The effect of interactions between conditions (CONT; SS), load (70%, 85%, and 100% 1RM), and muscle activity (triceps brachii, pectoralis major, and anterior deltoid) were assessed using a three-way 2 × 3 × 3 (condition × load × muscle) repeated measures analysis of variance (ANOVA). The post hoc comparisons were conducted using the Tukey’s test. The level of significance was set at *p* < 0.05 for all statistical analyses. Effect sizes and 95% confidence intervals were reported and presented. The effect sizes for main effects and interactions were determined by partial eta squared (η^2^) values. Partial eta squared (η^2^) values were classified as small (0.01 to 0.059), moderate (0.06 to 0.137), and large (>0.137). Effect size (ES) was determined for pairwise comparisons using Cohen’s d, and 95% confidence intervals were also calculated. ES was defined as large (d > 0.8), moderate (d between 0.79 and 0.5), small (d between 0.49 and 0.20), and trivial (d < 0.2) [37].

## 3. Results

The repeated measures three-way ANOVA was computed and showed a statistically significant main interaction for condition × muscle group (*p* < 0.01; η^2^ = 0.569); condition × load (*p* < 0.01; η^2^ = 0.418); load × muscle group (*p* < 0.01; η^2^ = 0.709; Table 1). A main effect also occurred for load (*p* < 0.01; η^2^ = 0.976); condition (*p* < 0.01; η^2^ = 0.968); and muscle group (*p* < 0.01; η^2^ = 0.977; Table 1). There was no statistically significant main multi interaction for condition × load × muscle (*p* = 0.47; η^2^ = 0.091). The post hoc tests for the main effect of condition revealed a statistically significant decrease in %MVIC for the SS compared to the CONT condition (74.9 vs. 88.9 %MVIC; *p* < 0.01; ES = 0.39). Further, the post hoc analyses for the main effect of load indicated a statistically significant increase in %MVIC for 100%1RM when compared to 85%1RM (95.7 vs. 79.7 %MVIC; *p* < 0.01; ES = 0.80), and to 70%1RM (95.7 vs. 79.7 %MVIC; *p* < 0.01; ES = 1.17), as well as a statistically significant increase in %MVIC for 85%1RM compared to 70%1RM (79.7 vs. 70.4 %MVIC; *p* < 0.01; ES = 0.39). Moreover, the post hoc analysis for the main interaction effect of condition × load demonstrated a statistically significant decrease in %MVIC for the SS condition compared to CONT at loads of 70%1RM (*p* < 0.01; ES = 0.42); 85%1RM (*p* < 0.01; ES = 0.67); and 100%1RM (*p* < 0.01; ES = 1.12; Table 2). Similarly, the post hoc analysis for the main interaction effect of condition × muscle revealed a statistically significant decrease in %MVIC for the SS condition in comparison to the CONT at all test loads for the anterior deltoid (*p* < 0.01; ES = 1.69), pectoralis major (*p* < 0.01; ES = 0.61), and triceps brachii (*p* < 0.01; ES = 1.23; Table 3). The post hoc analysis for the main interaction effect of load × muscle is presented in Table 4, and post hoc results for multi interaction are presented in Table 5.

## 4. Discussion

The main finding of the present study was a significantly lower peak muscle activity of the prime movers during the bench press using the SS device in comparison to CONT bench press conditions at the same absolute external loads. Moreover, our findings confirm that the external load and the use of the SS affect muscle activity considered during the bench press exercise.

To date, only two studies have examined the effect of the SS on muscle activity of the prime movers during the flat bench press movement [20,23]. A study by Ye et al. [23] indicated a significant increase in the mean barbell velocity and 1RM during bench press with the SS. The SS supportive device seems to have an acute effect on 1RM bench press performance. Using the SS reduces peak activity of muscles up to 1RM values, causing a decrease in the recruitment of motor units and the frequency of excitation [4]. This may explain a lower level of activity for all studied muscles during submaximal and maximal loading (70%, 85%, and 100% 1RM). The presented results indicate a decrease in activity of all the studied muscles during the bench press at the 1RM load with the assistance of the SS device, which was confirmed in previous research [20]. Dugdale et al. [20] studied the effects of the SS on barbell velocity and muscle activity in male powerlifters. Normalized activity for all muscles (grouped) was reduced in the SS trials. Moreover, the SS absorbs part of the external load by providing additional elastic enhancement, which reduces muscle tension. However, at the same time, the use of the SS during the bench press exercise allows for overcoming greater loads than without the use of this supportive device [20,21,23]. The authors suggest the possibility of a mechanically favorable position of the elbows and a shift of the sticking point as a possible explanation of this phenomenon [23]. Furthermore, research suggests that using the SS in training allows athletes to perform a greater volume of work, which may contribute to greater strength outcomes [21].

Changes in muscle activity after the application of the SS occurred both in the total and individual outcomes. It is noteworthy that the use of the SS at the same load caused an insignificant decline in muscle activity of the anterior deltoid. The authors point to a less significant role of the anterior deltoid in the bench press movement, which is not consistent with other bench press research, indicating the critical role of the anterior deltoid in overcoming maximum loads [9]. On the contrary, we found that the anterior deltoid showed the highest activity (%) in all conditions, and its activity decreased with the use of the SS at all loads. Thus, the changes in the activity of particular muscles due to the variation in external loads and conditions may be partially attributed to the lifting technique, different levels of muscular strength of these muscles, and previous injuries [38,39]. A decrease in muscle activity due to use of the SS can also be caused by reducing the requirements in the three-plane stabilization process during the bench press [8]. Research shows that a decrease in stabilization results in increased movement requirements and increases the activity of individual muscles [8,40,41], along with the loss of force output [42]. Considering the construction and usage of the SS, this seems to be caused by the increased stabilization and connection of the elbows with the layer of fabric. Thus, the SS can be used in the continuum of rehabilitation exercises based on muscle activation, or implemented among people with shoulder instability [22]. Moreover, it seems that use of the SS reduces the stress placed on the shoulder and elbow joints [23]. The conclusion of previous SS studies coincide with our results, and indicate the highest decrease in muscle activity of the triceps brachii while using this device. This is due to the greatest mechanical assistance at the beginning of the positive work of the bench press lift, where the triceps brachii shows the highest activity among the examined muscles [43]. It should be taken into account that the SS may have a negative effect on training outcomes by reducing muscle activity [44]. According to previous research, exercises producing higher levels of muscle activity are associated with greater long-term strength and hypertrophy adaptive changes [45]. The results of our research suggest that achieving similar muscle activity requires an additional 15% to the 1RM (70% 1RM CONT vs. 85% 1RM SS, 85% 1RM CONT vs. 100% 1RM SS; Table 2). Therefore, when programming a training protocol, one should not omit systematic progress of exercise intensity with an additional load when using the SS [46,47]. Importantly, comprehensive training should include triceps brachii accessory exercises due to the significantly diminished activity of this muscle while using the SS during the bench press exercise.

A significant increase in total %MVIC of the prime movers between subsequent loads that previous authors suggested [20] was also confirmed in our study (Table 4). This confirms previous research indicating the impact of increasing load on muscle and efferent motor activity [5,7,9]. It should be noted that with the use of lower external loads during exercises with the SS, muscle activity changes to a lesser extent. The difference in %MVIC of three muscle groups between the CONT and SS groups increased with the progression of the load up to 100% 1RM, where it amounted to 17% (Table 2). In regards to the increasing assistance of the SS device along with a decline in muscle activity, the results of the presented study showed that the most advisable use of the SS should start with the load exceeding particular %1RM, which will vary individually. With the load not exceeding this threshold, the SS effect may be negligible.

Fatigue can be an important factor affecting muscle activity, which should be taken into account in research protocols [48]. The only study that measured muscle activity with the SS on submaximal loads of 70% and 87.5% 1RM [20] was associated with high fatigue levels by time-consuming procedures with multiple repetitions. Protocols used in studies can significantly affect sEMG measurements, with particular regard to the pre-exhaustion and no-exhaustion methods [38]. Moreover, introducing pre-exhaustion exercise before the bench press may lead to an increased triceps brachii and decreased pectoralis major activity [49]. While such activities can both increase and decrease the activity of the muscles, multiple repetitions cause increased prime movers activity during the bench press exercise [25]. Taking this into consideration, we have limited the attempts in our procedure to determine the actual impact of the SS with minimum fatigue. Some authors suggest that fatigue can be a protective strategy to maintain muscle reserve and retrain muscle activity in case of muscle damage [49]. Furthermore, low fatigue resistance training seems to stimulate strength gains as much as high fatigue protocols while minimizing training discomfort [50]. A low number of repetitions used in our exercise protocol in combination with high intensity is used both in training focused on the development of power and maximum strength among competitive powerlifters [19]. The results of our research indicate a similar influence of the SS in low fatigue conditions as well as in more exhausting procedures [20], with the greatest decreases in anterior deltoid and triceps brachii muscle activity. This indicates the possibility of using the SS in strength and power training based on a low number of repetitions and high velocity without significant fatigue [26,51,52].

The present study has several limitations that should be addressed. Although the results showed significant changes in activity of the prime movers between the bench press with and without the SS device at different absolute external loads, the muscle activity measurements were conducted only from one side of the body (dominant), and the reliability of the MVIC was not measured. Moreover, the evaluation of the external structure of the movement (i.e., forces and movement torques) was omitted in this study, nor were the kinematics evaluated of the two bench press conditions. Furthermore, relative loads were used rather than absolute loads. Future research should consider the evaluation of the SS device on activity of the prime movers from both sides of the body, as well as chronic effects of the SS in relation to strength, power, and muscular hypertrophy.

## 5. Conclusions

The presented study compared the activity of selected muscles during conventional and SS-supported bench press at 70%, 85%, and 100% 1RM among resistance-trained men. Considering the results of our study, it can be stated that the SS elastic device significantly affected the muscle activity pattern of the flat bench press exercise. The use of the SS resulted in a decrease in activity of the triceps brachii, pectoralis major, and anterior deltoid muscles during the flat bench press in comparison to lifting an equal load under standard conditions. Moreover, muscle activity decreased along with the assistance of the SS device, which was greater at higher loads. The SS device may be an effective tool to reduce the activity of chosen muscles and increase stability during movement, which potentially can be beneficial during the rehabilitation process and ensure a faster return to training. The influence of the SS has also been confirmed in the absence of increasing fatigue within the sets, allowing implementation of the device in specific strength and power programs. However, considering the decreased activity of the prime movers, additional load and supplemental exercises should be included in training protocols focused on strength and hypertrophy.

## Figures and Tables

**Figure 1 ijerph-17-07695-f001:**
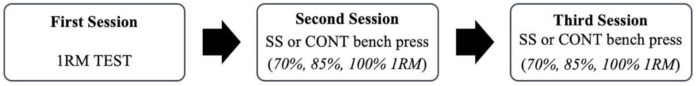
Schematic representation of the experimental sessions protocol.

**Figure 2 ijerph-17-07695-f002:**
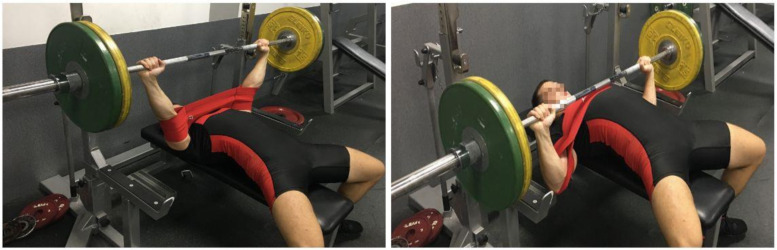
SS placement during a sample flat bench press repetition. Participants were instructed to wear the SS sleeve in the middle of the elbows.

**Table 1 ijerph-17-07695-t001:** The results of a three-way repeated measures ANOVA.

ANOVA (Number of Factors)	F	*p*	η^2^
Condition (2)	273.5	0.01 *	0.968
Load (3)	359.5	0.01 *	0.976
Muscle (3)	386.3	0.01 *	0.977
Condition × Load (2 × 3)	6.5	0.01 *	0.418
Condition × Muscle (2 × 3)	11.9	0.01 *	0.569
Load × Muscle (3 × 3)	22.0	0.01 *	0.709
Condition × Load × Muscle (2 × 3 × 3)	0.9	0.47	0.091

* statistically significant differences at *p* < 0.05.

**Table 2 ijerph-17-07695-t002:** Post hoc analysis for the main interaction of two conditions vs. three loads.

Load	%MVIC of 3 Muscle Groups for CONT	%MVIC of 3 Muscle Groups for SS	*p*	ES
70%1RM	75.6 ± 26.0	65.1 ± 23.5	0.001 *	0.42
85%1RM	86.9 ± 21.7	72.4 ± 21.5	0.001 *	0.67
100%1RM	104.2 ± 16.0	87.2 ± 14.4	0.001 *	1.12

Data are presented as mean ± standard deviation; * statistically significant differences at *p* < 0.05.

**Table 3 ijerph-17-07695-t003:** Post hoc analysis for the main interaction of two conditions vs. three muscle groups.

Muscle Group	%MVIC of 3 Loads for CONT	%MVIC of 3 Loads for SS	*p*	ES
Anterior deltoid	115.0 ± 9.8	100.5 ± 7.1	0.001 *	1.69
Pectoralis major	67.0 ± 17.0	57.8 ± 13.0	0.001 *	0.61
Triceps brachii	84.7 ± 15.2	66.5 ± 14.5	0.001 *	1.23

Data are presented as mean ± standard deviation; * statistically significant differences at *p* < 0.05.

**Table 4 ijerph-17-07695-t004:** Post hoc analysis for the main interaction of three loads vs. three muscle groups.

Load	Anterior Deltoid %MVIC	
70%1RM	102.1 ± 9.9	
85%1RM	107.9 ± 10.3	
100%1RM	113.4 ± 10.6	
	***p***	**ES**
70%1RM vs. 85%1RM	0.135	0.57
70%1RM vs. 100%1RM	0.001 *	1.10
85%1RM vs. 100%1RM	0.182	0.53
**Load**	**Pectoralis Major %MVIC**	
70%1RM	45.9 ± 5.4	
85%1RM	61.5 ± 7.0	
100%1RM	79.8 ± 8.9	
	***p***	**ES**
70%1RM vs. 85%1RM	0.001 *	2.50
70%1RM vs. 100%1RM	0.001 *	4.61
85%1RM vs. 100%1RM	0.001 *	2.29
**Load**	**Triceps Brachii %MVIC**	
70%1RM	63.2 ± 9.6	
85%1RM	69.6 ± 11.9	
100%1RM	94.0 ± 12.0	
	***p***	**ES**
70%1RM vs. 85%1RM	0.067	0.59
70%1RM vs. 100%1RM	0.001 *	2.83
85%1RM vs. 100%1RM	0.001 *	2.04

Data are presented as mean ± standard deviation; * statistically significant differences at *p* < 0.05.

**Table 5 ijerph-17-07695-t005:** Results of the main multi interaction effect of condition × load × muscle group.

Muscle Group	%MVIC for CONT (95% CI)	%MVIC for SS (95% CI)	*p*	ES
70% 1RM				
Anterior deltoid	108.0 ± 8.8	96.1 ± 7.1	0.001 *	1.49
	(101.7 to 114.3)	(91.0 to 101.2)		
Pectoralis major	48.1 ± 5.6	43.7 ± 4.4	0.755	0.87
	(44.1 to 52.1)	(40.6 to 46.8)		
Triceps brachii	70.7 ± 6.6	55.6 ± 5.1	0.001 *	2.56
	(66.0 to 75.4)	(52.0 to 59.2)		
85%1RM				
Anterior deltoid	114.6 ± 8.1	101.1 ± 7.6	0.001 *	1.72
	(108.8 to 120.4)	(95.7 to 106.5)		
Pectoralis major	66.4 ± 5.6	56.6 ± 4.5	0.001 *	1.93
	(62.4 to 70.4)	(53.4 to 59.8)		
Triceps brachii	79.8 ± 6.1	59.4 ± 5.6	0.001 *	3.48
	(75.4 to 84.2)	(55.4 to 63.4)		
100%1RM				
Anterior deltoid	122.5 ± 6.0	104.2 ± 4.0	0.001 *	3.59
	(118.2 to 126.8)	(101.3 to 107.1)		
Pectoralis major	86.6 ± 6.3	73.0 ± 4.8	0.001 *	2.43
	(82.1 to 91.1)	(69.6 to 76.4)		
Triceps brachii	103.5 ± 5.5	84.4 ± 8.6	0.001 *	2.65
	(99.6 to 107.4)	(78.3 to 90.5)		

Data are presented as mean ± standard deviation; * statistically significant differences at *p* < 0.05; CI = confidence interval.

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
