# Peer review of "Impact of the “Sling Shot” Supportive Device on Upper-Body Neuromuscular Activity during the Bench Press Exercise"

_ijerph, 2020, doi:10.3390/ijerph17207695_

Round 1

Reviewer 1 Report

The work is well presented. Its structure and statistical design are good.

However, it presents two important problems, one in the objective and the other in the methodology.

The objective proposed by the authors does not seem to have scientific relevance. A priori it seems obvious that the use of a device that helps to reduce the load on the shoulder musculature in the bench press movement reduces the electromyographic activity of its musculature.

The authors should have carried out a study of the repeatability of the maximum peaks of the sEMG of each muscle, in each of the conditions and with the different loads, in order to know if the values of the proposed measurement tests are a measure stable enough to be studied (Hopkins WG (2015) Spreadsheets for analysis of validity and reliability. Sportscience 19: 36-42)

Title

Must be modified

The noun acute is a qualitative assessment that does not fit in quantitative research studies.

Neuromuscular activity is not from the bench press but from the shoulder muscles.

I suggest

The effects of “Sling Shot” supportive device for bench press on shoulder muscles neuromuscular activity.

Introduction

In the introduction it is not clear why the objective of the study is necessary. The need to study the sEMG of the shoulder musculature using the SS device should be justified

The word “bench press” first appears on line 31, and the use of your initials (BP) appears on line 98 without reference to which term it refers to. This must be corrected.

Methodology

The authors should explain in more detail how they randomize the sample and what the crossover design consisted of.

It must be explained whether the order of the experimental sessions was the same for all (first with SS and second without SS) or was it in a random order.

Given that the action analyzed is very specific and the device used is the objective to be evaluated, I consider that the deformation value of the material used should be provided under a constant load.

The timing of the video and electromyographic signal should be explained in more detail.

Results

The results in Table 1 and those described in lines 174-180 do not include effect size values. They must be included in all cases.

Table 3 has an error in the first column. You must change the word load to muscle groups.

Why aren't post hoc results shown for load x muscle?

Table 4 shows the interaction of load x condition x muscle, however, table 1 already indicates that there is no effect of the interaction of these three factors. Why is the data in this table displayed?

Discussion

The most obvious explanation for the study result is not given. The significant decrease in peak muscle activity is due to the reduction of the load to be mobilized by the help of SS.

Does the use of SS not modify the real load moved? The answer to this question should contribute to answering statements used in the discussion such as “SS provides additional elastic tension, reducing muscle tension but at the same time allowing to overcome greater loads”, or as “using the SS in training allows athletes to perform a greater volume of work "

Line 238, It must be indicated which table of the results they refer to.

Conclusions

The text in this section contains information that is not directly derived from the results of the study carried out, so the entire section should be reviewed.

Reviewer 2 Report

 This peer-reviewed paper has the following major problems, and it is inappropriate to publish it as the original paper (article). It may be better to reformat the form and repost as communication etc.

  1. Kinematic measurements have not been performed

 In this type of research, kinematic measurements that indicate what kind of motion was performed are essential. Then, it is necessary to analyze the electromyogram in accordance with the kinematic diagram of the bench press movement. Analysis of EMG in this type of study is meaningless unless the range of bench press motions of interest is clear. The analyzed phase of motion was not indicated at all.

  1. It is not clear whether the purpose of using SS is to increase the external load, or the physiological load, or both.

 The decrease in muscle activity for a certain external load means a decrease in physiological load (stimulus). This is also known from previous studies. In addition, the physiological meaning of increasing the maximum external load by using SS is not clarified.

  1. When using abbreviations, write the official name in the text

  1. There is no data to support the following statements in the abstract or conclusion.

 "it can be stated that the SS elastic device significantly affected the movement structure of the flat bench press.''

 "The results of this study showed that using the SS significantly affects the movement structure of the flat bench press and results in an acute decrease of muscle activity during the flat bench press in comparison to an equal load under CONT conditions.

5. As mentioned above, a decrease in muscle activity is recognized as a decrease in physiological stimulus. Therefore, it is necessary to reconsider the meanings of the following description in the conclusion. In addition, it is unclear how the phase was identified in the absence of kinematic measurements.

 "The SS device may be an effective tool to overload selected bench press phases with a reduction of certain muscle activity.''

Reviewer 3 Report

Major comments

Introduction requires a stronger rationale for why this matters. You have some nice data showing that muscle activation is lower with sling shot  but why does this matter? Could it be used for rehabilitation? You can't claim it's for overload because you're overloading less, unless you reference some of Stuart Phillips' work showing that you can do higher reps at lower loads to increase muscle protein synthesis. 

You must provide references to show some of the methods you use (e.g. for testing isometric strength or the warm up protocols) have been validated

Minor comments

Line 45-46 does not make sense; needs rephrasing

Line 92 - why was the Declaration of the WMA used as opposed to the Declaration of Helsinki that is standard for non-US based research? 

Line 94-104 - we know circadian rhythms have a huge effect on strength. Were all sessions conducted at the same time of day for each participant and between participants?

Line 107 - 109 - you talk about the size of SS used. Do these provide different amounts of support (tension) based on size? Perhaps in the supplemental materials it may be useful to have a breakdown of individual subjects' anthropometry and which band was used and ratios of their unassisted 1RM to the tension provided by the SS they used. Having a summary sentence to address this in the manuscript might also be useful.

Line 110 - noted you address above. Might be worth adding in a short sentence in lines 94-104 before it comes up at 110 - seems like a natural place to have this info

Line 111-112 - is this a standard warm up procedure? Validated or previously used in published literature? Reference of validation required.

Line 115-116 - is this standard procedure? Might be worth adding a little more detail on where this protocol comes from and why the increments

Line 134 Figure 1 - might be worth showing an image with clips on the end of the plates to show normal health and safety has been followed during the research

Line 155 - 159 - please provide a reference to show this is a validated method for assessing isometric strength. Why has an isokinetic dynamometer not been used as gold standard? If you have not had access to one then you need to provide evidence to show this method is valid

Line 161- details of statistics software - place, manufacturer etc

Line 170 - new page

Discussion - same as for the introduction. There needs to be a stronger explanation of why this matters. Could you talk about prevalence of use or injury rates with slingshot? 

Round 2

Reviewer 1 Report

The authors have attended most of the suggestions made in the first review. However, as I explained in the first review, the scientific relevance of the study is unclear and remains unclear.

The key is in the justification of the need for the use of SS. It is obvious that the use of the device reduces the physiological load loading the same external load. The potential interest of the study is not that it allows working longer with submaximal external loads and even with loads higher than its maximum load. This result is a logical consequence of the help provided by the use of the SS device. What would be of interest is to know the consequences of working with this difference between physiological load and external load using SS, but this has not been the approach of the work. If the advantage is self-confidence, the study should have been considered to evaluate it. If the advantage is that it is possible to work with supramaximal external loads, the effect of this type of advantage should have been considered.

Author Response

The authors have attended most of the suggestions made in the first review. However, as I explained in the first review, the scientific relevance of the study is unclear and remains unclear.

The key is in the justification of the need for the use of SS. It is obvious that the use of the device reduces the physiological load loading the same external load. The potential interest of the study is not that it allows working longer with submaximal external loads and even with loads higher than its maximum load. This result is a logical consequence of the help provided by the use of the SS device. What would be of interest is to know the consequences of working with this difference between physiological load and external load using SS, but this has not been the approach of the work. If the advantage is self-confidence, the study should have been considered to evaluate it. If the advantage is that it is possible to work with supramaximal external loads, the effect of this type of advantage should have been considered.

Reply: We agree with the reviewer's opinion and we will certainly include it in future research related to the use of the Sling shot device. It should be taken into account that our research is one of the first to address the subject of the SS and should be extended in the future to comprehensively describe the subject matter. We would like to emphasize that the study aimed to identify changes in the pattern of muscle activity between bench press attempts with and without the SS device for selected external loads taking into account work on a low level of fatigue conditioned by the research procedure. The results showed that the support obtained with the Sling shot device is not the same for every external load and differs between muscles. Using the SS does not unload each muscle to the same degree, causing the pattern of muscle activity to change along with the external load. The aspect of stabilization during the bench press exercise was also discussed, which did not appear in earlier works concerning the use of the SS. We believe our study provides valuable guidelines for power and strength training programs regarding the use of the Sling shot.

Minor changes to the text and formatting were adopted following this review. All the revisions are highlighted using the “Track changes” function in Microsoft Word, marked in red.

Reviewer 2 Report

 Authors are needed to re-examine the experimental procedures necessary for the internal validity, rational and logical consideration of the obtained findings

Author Response

Authors are needed to re-examine the experimental procedures necessary for the internal validity, rational and logical consideration of the obtained findings.

Reply: We agree with the reviewer's opinion and we will certainly include it in future research related to the use of the Sling shot device. It should be taken into account that our research is one of the first to address the subject of the SS and should be extended in the future to comprehensively describe the subject matter. We would like to emphasize that the study aimed to identify changes in the pattern of muscle activity between bench press attempts with and without the SS device for selected external loads taking into account work on a low level of fatigue conditioned by the research procedure. The results showed that the support obtained with the Sling shot device is not the same for every external load and differs between muscles. Using the SS does not unload each muscle to the same degree, causing the pattern of muscle activity to change along with the external load. The aspect of stabilization during the bench press exercise was also discussed, which did not appear in earlier works concerning the usage of the SS. We believe our study provides valuable guidelines for power and strength training programs regarding the use of the Sling shot.

Minor changes to the text and formatting were adopted following this review. All the revisions are highlighted using the “Track changes” function in Microsoft Word, marked in red.